# Recognition of a highly conserved glycoprotein B epitope by a bivalent antibody neutralizing HCMV at a post-attachment step

Xiaohua Ye[1]☉, Hang Su[1,2]☉, Daniel Wrapp[3]☉, Daniel C. Freed[4], Fengsheng Li[4], Zihao Yuan[1], Aimin Tang[4], Leike Li[1], Zhiqiang Ku[1], Wei Xiong[1], Dabbu Jaijyan[5], Hua Zhu[5], Dai Wang[4], Jason S. McLellan[3], Ningyan Zhang[1]*, Tong-Ming Fu[1,4]*, Zhiqiang An[1]*

1 Texas Therapeutics Institute, Brown Foundation Institute of Molecular Medicine, University of Texas Health Science Center at Houston, Houston, Texas, United States of America, 2 Wuya College of Innovation, Shenyang Pharmaceutical University, Shenyang, China, 3 Department of Molecular Biosciences, University of Texas at Austin, Austin, Texas, United States of America, 4 Merck Research Laboratory, Merck & Co., Inc., Kenilworth, New Jersey, United States of America, 5 Department of Microbiology, Biochemistry and Molecular Genetics, Rutgers New Jersey Medical School, Newark, New Jersey, United States of America

☉ These authors contributed equally to this work.
* ningyan.zhang@uth.tmc.edu (NZ); Tong-Ming.Fu@uth.tmc.edu (TMF); zhiqiang.an@uth.tmc.edu (ZA)

**Data Availability Statement:** All relevant data are within the manuscript and its Supporting Information files.

## Abstract

Human cytomegalovirus (HCMV) is one of the main causative agents of congenital viral infection in neonates. HCMV infection also causes serious morbidity and mortality among organ transplant patients. Glycoprotein B (gB) is a major target for HCMV neutralizing antibodies, yet the underlying neutralization mechanisms remain largely unknown. Here we report that 3–25, a gB-specific monoclonal antibody previously isolated from a healthy HCMV-positive donor, efficiently neutralized 14 HCMV strains in both ARPE-19 cells and MRC-5 cells. The core epitope of 3–25 was mapped to a highly conserved linear epitope on antigenic domain 2 (AD-2) of gB. A 1.8 Å crystal structure of 3–25 Fab in complex with the peptide epitope revealed the molecular determinants of 3–25 binding to gB at atomic resolution. Negative-staining electron microscopy (EM) 3D reconstruction of 3–25 Fab in complex with de-glycosylated postfusion gB showed that 3–25 Fab fully occupied the gB trimer at the N-terminus with flexible binding angles. Functionally, 3–25 efficiently inhibited HCMV infection at a post-attachment step by interfering with viral membrane fusion, and restricted post-infection viral spreading in ARPE-19 cells. Interestingly, bivalency was required for HCMV neutralization by AD-2 specific antibody 3–25 but not the AD-4 specific antibody LJP538. In contrast, bivalency was not required for HCMV binding by both antibodies. Taken together, our results reveal the structural basis of gB recognition by 3–25 and demonstrate that inhibition of viral membrane fusion and a requirement of bivalency may be common for gB AD-2 specific neutralizing antibody.

**Funding:** Z.A received all the grants including grants from Merck & Co., the Texas Emerging Technology Fund, and the Welch Foundation Grant No. AU-0042-20030616. D.F, F.L, A.T and T.F were employees of Merck & Co. when the study was performed. D.F, F.L and A.T participated data collection and analysis. T. F participated study design, data analysis and manuscript preparation.

**Competing interests:** I have read the journal's policy and the authors of this manuscript have the following competing interest: Tong-Ming Fu and Zhiqiang An filed a patent on 3-25 antibody.

## Author summary

HCMV infection is usually asymptomatic in healthy individuals. However, life-threatening diseases frequently accompany HCMV infection in individuals with under-developed or compromised immune systems. Glycoprotein B antigenic domain 2 (AD-2) is a major target for HCMV-neutralizing antibodies that potentially provide immune protection. We report the structure-based study of gB recognition by a potent neutralizing antibody named 3–25 that binds a highly conserved epitope on AD-2. Functionally, 3–25 efficiently inhibited HCMV infection at a post-attachment step by interfering with viral membrane fusion, and restricted post-infection viral spreading. Furthermore, bivalency of 3–25 is required for viral neutralization but not for binding. Our findings advance understanding of gB antibody-mediated HCMV neutralization and facilitate development of gB-targeted vaccines and antibody drugs against HCMV infection.

## Introduction

Human cytomegalovirus (HCMV), also known as cytomegalovirus (CMV) or human herpesvirus 5 (HHV5), is a β-herpesvirus that causes life-long infection in humans of all ages [1]. The seroprevalence of HCMV infection varies between 40–100% among different human populations [2]. HCMV is also one of the most common congenital viral infections that cause permanent damage to the developing central nervous systems of infants [3–5]. HCMV infection usually causes mild or unnoticeable symptoms in healthy adults. However, primary HCMV infection or reactivation may cause life-threatening diseases in AIDS patients or organ transplant recipients [6, 7]. Effective HCMV vaccines and potent antiviral drugs are thus highly desirable [8].

HCMV has an internal icosahedral capsid that encloses a double-stranded DNA genome (~235 kb), a mostly disordered tegument layer in-between, and an external membrane decorated with glycoprotein complexes [1, 9]. Over 250 open reading frames (ORFs) have been identified in the genome of clinical HCMV isolates [10]. The functions of most HCMV-encoded proteins are unknown except some well-characterized viral proteins, including gB, gH, gL, gO, pUL128, pUL130, and pUL131, which play major roles in viral infection. The gH/gL/gO trimeric complex interacts with receptor PDGFRα and is required for HCMV infection of all cell types [11–13]. The gH/gL/pUL128-131 pentameric complex, which engages receptors NRP-2 and OR14I1, is essential for HCMV infection of epithelial cells, endothelial cells, and myeloid cells [14–16]. The HCMV gB protein (gpUL55) is a type III viral fusion protein that shares structural similarities with gB proteins of herpes simplex virus 1 (HSV-1) and Epstein-Barr virus (EBV) [17]. The gB protein is absolutely required for HCMV cellular entry and cell-to-cell spreading [18]. Although gB was reported to bind EGFR and PDGFRα [12, 19], there is also report that gB promoted HCMV entry *in trans* as a viral fusion protein rather than a receptor-binding protein [20]. Monoclonal antibodies targeting different HCMV glycoproteins were isolated from HCMV seropositive donors or vaccinated animals and comprehensively characterized [21–23]. Generally, the pentamer UL specific antibodies are extremely potent neutralizers in epithelial cells, endothelial cells, and myeloid cells but show no neutralizing effect in fibroblast cells. The gB- and gH-specific antibodies show broad neutralization in both epithelial and fibroblasts cells but with much lower potency than antibodies specific for the pentamer UL proteins [21].

Because of its essential role in viral infection, gB has been a major target for development of HCMV vaccines and antiviral drugs. Two gB-based vaccines, gB/MF59 (Sanofi) and ASP0113

(Vical, Astellas), and two gB antibodies, LJP538 (Novartis) and TCN202 (Theraclone), have been tested in clinical trials [24, 25]. The gB-specific antibodies in CMV-infected individuals target five major antigenic domains (ADs) [26]. The antigenic domain 2 (AD-2), located at the N-terminus of gB, is one of the major antigenic domains targeted by gB-specific antibodies isolated from CMV-infected individuals [26]. AD-2 contains a highly conserved site I epitope (amino acids 68–77) that is targeted by neutralizing antibodies and a strain-specific site II epitope (amino acids 50–54) that is targeted by non-neutralizing antibodies [27]. Only about 50% of human sera from HCMV-infected donors recognize gB AD-2 [28], suggesting that AD-2 is not immunodominant. Several studies implicate gB AD-2 specific antibodies as a correlate of protective immunity against HCMV infection or disease. Lack of antibodies against gB AD-2 is associated with CMV disease after renal transplantation in recipients having the same glycoprotein H serotypes as their donors [29]. A decreased incidence of viremia was correlated with higher antibody levels against gB AD-2 but not with antibody levels against the other three ADs (AD-1, AD-4 and AD-5) among gB/MF59 vaccinated seropositive solid organ transplant recipients [30]. The magnitude of maternal AD-2 specific antibodies was borderline associated with low risk of congenital CMV infection among HIV-1 exposed infants [31]. Despite the importance of gB AD-2, little is known about the neutralization mechanism of gB AD-2 specific antibodies.

We previously isolated a panel of monoclonal antibodies from three HCMV seropositive donors. One antibody named 3–25 showed the most potent virus neutralizing ability among gB-specific antibodies (23). In this study, we evaluated the neutralization efficacy of 3–25 against infection of 14 HCMV strains and mapped the epitope of 3–25 to a highly conserved site on AD-2 of gB. We resolved a 1.8 Å crystal structure of 3–25 Fab plus epitope peptide that revealed the molecular determinants of 3–25 binding to gB at atomic resolution. Our negative-stain electron microscopy (EM) 3D reconstruction of 3–25 Fab in complex with de-glycosylated postfusion gB showed that 3–25 Fab fully occupied the gB trimer at the N-terminus with flexible binding angles. Functionally, the 3–25 antibody efficiently inhibited HCMV infection at a post-attachment step by interfering with viral membrane fusion and restricted post-infection viral spreading in ARPE-19 cells. Interestingly, bivalency of 3–25 antibody was required for virus neutralization but not for binding, which suggested a bivalent-binding-dependent neutralization mechanism of 3–25. Our studies suggest that gB AD-2 specific neutralizing antibodies require bivalency and function through inhibition of viral membrane fusion. Our findings shed light on the mechanism of gB antibody-mediated HCMV neutralization.

## Results

### Neutralization potency and breadth of 3–25

We reported previously that 3–25 bound recombinant gB and HCMV virions in an ELISA assay, and neutralized infection of AD169 strain [23]. Here, twelve previously described HCMV clinical isolates [32] together with two pentamer-restored laboratory-adapted strains (Towne-ts15-rR and AD169rev) were used to evaluate the *in vitro* neutralization potency and breadth of 3–25 in ARPE-19 epithelial cells and MRC-5 fibroblast cells. HCMV hyperimmune globulin (HIG, also called CytoGam), which is a CMV prophylacic used clinically [33], was included as a positive control. As shown in **Fig 1A and 1B,** 3–25 neutralized infection of all tested strains in both ARPE-19 cells and MRC-5 cells. Notably, the IC$_{50}$ of 3–25 and CytoGam in two cell lines and for all viral strains were 15–188.3 ng/ml and 457–34296 ng/ml (**S1 Fig, S1 Table**) respectively, suggesting that 3–25 is much more potently neutralizing than CytoGam. Antibody 3–25 showed potent neutralization against a panel of clinical HCMV isolates in both

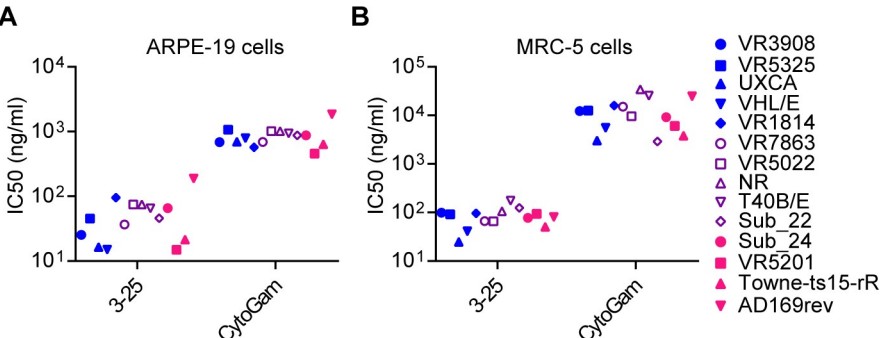

**Fig 1. Neutralization efficacy of 3–25 and CytoGam against a group of HCMV strains.** Twelve clinical HCMV isolates and two laboratory-adapted HCMV strains were used for neutralization assays in (**A**) ARPE-19 cells and (**B**) MRC-5 cells. The $IC_{50}$ values were calculated by non-linear fit of the percentage of virus inhibition vs. concentration (ng/mL). The curves were shown in **S1 Fig** and the $IC_{50}$ values are listed in **S1 Table**.

epithelial cells and fibroblasts, which is consistent with broad neutralization activity of gB-specific neutralizing antibodies [24].

## Epitope mapping of 3–25

A Western blot assay was performed to see whether 3–25 recognizes denatured gB protein with a 6×His tag. As shown in **Fig 2A**, 3–25 specifically recognized gB but not HCMV pp65 with a 6×His tag, while the anti-His tag antibody recognized both gB and pp65 (**Fig 2A**), suggesting that 3–25 binds gB on a linear epitope. Thus, a 15-mer peptide library (11 amino acids overlap) covering the gB extracellular domain of AD169 strain was screened to locate the epitope. Antibody 3–25 strongly reacted with two adjacent peptides spanning residues 65–83 of gB (**Fig 2B**), which maps to a highly conserved site I epitope of antigenic domain 2 (AD-2) [26]. Nine biotinylated peptides, which cover amino acids 59–88 of gB and carry different lengths of truncations, were titrated for 3–25 binding to determine the boundary of the epitope. As shown in **Fig 2C**, peptides with truncations N-terminal to Glu69 or C-terminal to Tyr78 showed dramatically reduced binding to 3–25, which narrowed down the core epitope of 3–25 to residues 69–78 of gB. Thirteen biotinylated 20-mer peptides (residues 64–83 of gB) that carry single alanine substitutions were titrated to locate potential critical residues for 3–25 binding. As shown in **Fig 2D**, the relative binding potency of peptides with single alanine substitution at residues Glu69, Ile71, Tyr72, Thr75 and Leu76 to 3–25 were reduced more than 10-fold; the relative binding potency of peptides with single alanine substitution at residues Thr70, Lys77 and Tyr78 to 3–25 were reduced about 5-fold. In contrast, alanine substitution at residues Asn68, Asn73, Thr74, Gly79 and Asp80 did not reduce but rather increased the binding potency of these peptides (**Fig 2D**). Together, these results reveal the core epitope and potential critical amino acids for 3–25 binding to gB.

Examination of gB sequences of HCMV clinical isolates in **Fig 1** revealed identical amino acid sequences at the core epitope of 3–25 (69-ETIYNTTLKY-78) (**S2 Table**). To determine sequence variance of the 3–25 binding site, a total of 317 HCMV gB protein sequences were retrieved from a virus pathogen database and analysis resource (ViPR) [34] and aligned around the AD-2 region (residues 59–88 of gB). A total of 266 sequences with available AD-2 regions were analyzed and the frequencies of unique sequences were summarized. Nine unique sequences were identified for amino acids 59–88 of gB (**Fig 2E**). The top two sequences account for over 93.5% of analyzed sequences. Interestingly, eight of the nine unique sequences are identical at the 3–25 core epitope (residues 69–78). The last sequence has a

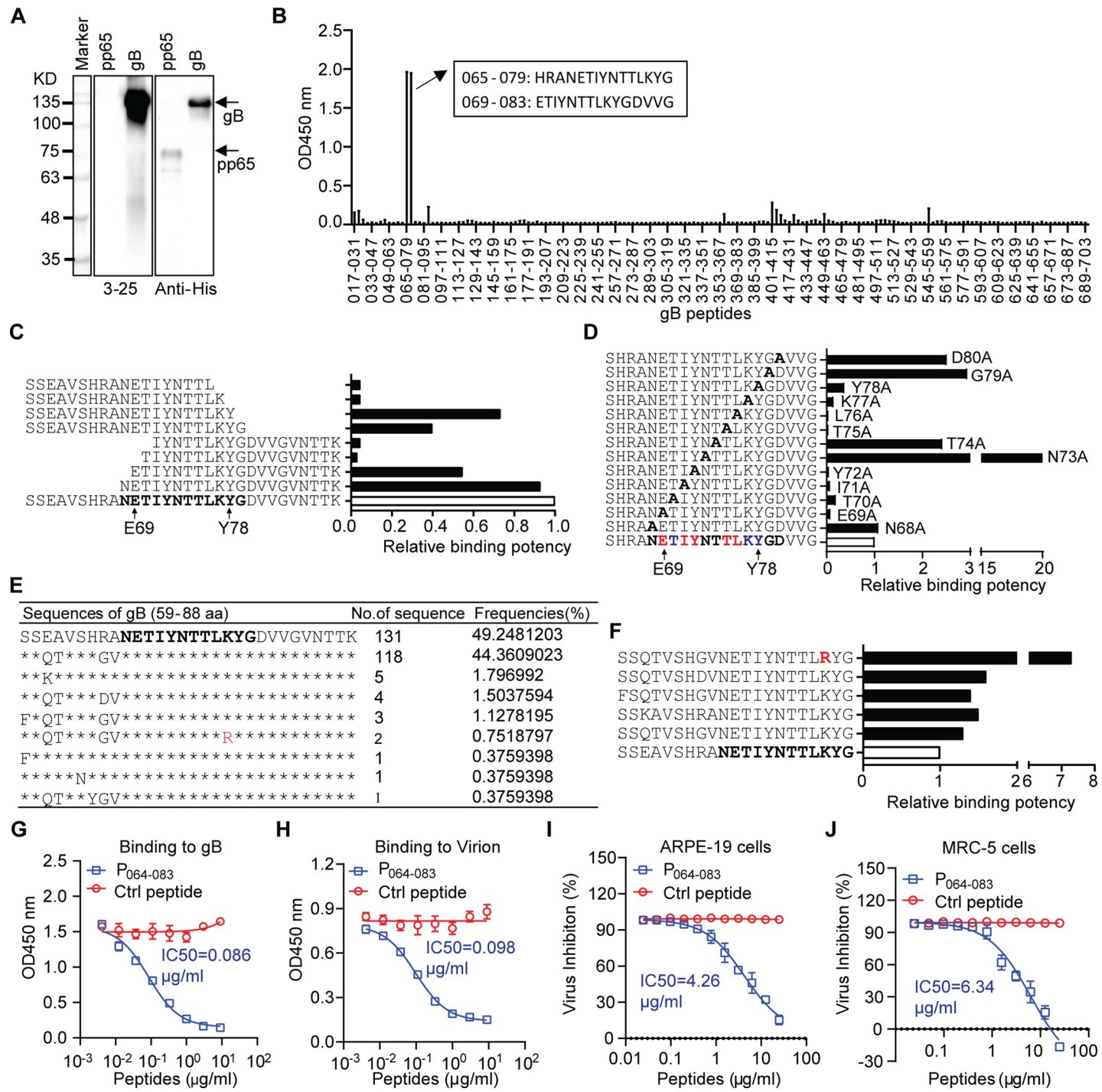

**Fig 2. Epitope mapping of 3–25.** (**A**) Detection of recombinant gB with 6×His tag using 3–25 and anti-His tag antibody by Western blot assay. Recombinant pp65 with 6×His tag served as control. (**B**) gB peptide library screening by ELISA assay. The binding of 3–25 to single gB peptides coated on 96-well plate was detected by HRP-conjugated anti-human IgG antibody. The amino acid sequences of two adjacent peptides with dominant binding to 3–25 are shown in a black box. (**C**) Biotinylated oligo peptides (residues 59–88 of gB) with different truncation lengths were titrated for 3–25 binding by ELISA assay. Relative binding potency was a reversed ratio of the $EC_{50}$ of truncated peptide to the $EC_{50}$ of full-length peptide (white column). The core 3–25 epitope is shown in bold. (**D**) Biotinylated epitope peptides with alanine substitutions were titrated for 3–25 binding by ELISA assay. Relative binding potency was a reversed ratio of the $EC_{50}$ of mutated peptide to the $EC_{50}$ of peptide with no mutation (white column). The impact of alanine substitution on binding to 3–25 was color coded at the bottom: bold, no reduced binding; blue, partially reduced binding; red, near abolished binding. (**E**) Analysis of HCMV gB sequences around 3–25 epitope. Identical amino acids to the sequence of the highest frequency are denoted with "*". A "K" to "R" substitution is shown in Red. (**F**) Biotinylated oligo peptides with representative gB sequences in Fig 2F were titrated for 3–25 binding $EC_{50}$ by ELISA assay. Relative binding potency was a reversed ratio of the $EC_{50}$ of a peptide to the $EC_{50}$ of the peptide with highest frequency (white column). (**G-H**)

Inhibition of 3–25 binding to (**G**) recombinant gB protein or (**H**) HCMV whole virion by pre-incubation with different concentrations of 3–25 epitope peptide P$_{064-083}$ or a non-binding control gB peptide. (**I-J**) Inhibition of 3–25 to neutralize AD169rev-GFP infection of (**I**) ARPE-19 cells or (**J**) MRC-5 cells by pre-incubation with different concentrations of 3–25 epitope peptide P$_{064-083}$ or a non-binding control gB peptide.

K77R substitution at the 3–25 core epitope, which accounts for 0.75% of analyzed sequences. Six biotinylated peptides representing sequence diversity around the 3–25 binding site were assessed for 3–25 binding. As shown in **Fig 2F**, all five peptides showed higher relative binding potency compared to the standard peptide. The peptide with the K77R substitution showed the highest binding potency, suggesting that 3–25 should recognize and neutralize HCMV strains with arginine at this position. These results demonstrated that 3–25 recognizes a highly conserved epitope among HCMV strains. The prevalence of 3-25-epitope-specific antibodies in serum samples of HCMV seropositive individuals was determined by ELISA assay. As expected, all three HCMV seronegative individuals showed negative reactions. Two of nine HCMV seropositive individuals (22.2%) showed positive reactions to the 3–25 epitope peptide (**S2A Fig**), which is lower in prevalence compared with reported 50% prevalence of whole AD-2 region specific antibodies [28].

To validate the epitope of 3–25, we determined whether an epitope peptide, named gB P$_{064-083}$ (64-SHRANETIYNTTLKYGDVVG-83), blocks the binding and neutralization ability of 3–25. Pre-incubation with gB P$_{064-083}$ but not a non-binding control gB peptide (57-VTSSEAVSHRANETI-71) inhibited the binding of 3–25 to gB protein and whole virions in a dose-dependent manner (**Fig 2G and 2H**). Presence of high concentrations of gB P$_{064-083}$ almost abolished the binding of 3–25 to both gB protein and whole virions. Similarly, presence of gB P$_{064-083}$ but not the control peptide blocked the neutralizing activity of 3–25 in ARPE-19 cells and MRC-5 cells in a dose-dependent manner (**Fig 2I and 2J**). Meanwhile, pre-incubation with gB P$_{064-083}$ alone showed no HCMV inhibiting effect (**S2B Fig**). These results demonstrate that the 3–25 epitope peptide specifically blocks the binding and neutralizing ability of 3–25.

## Structural basis for recognition of gB by 3–25

To further characterize the binding of 3–25 to gB, we performed negative stain electron microscopy on a complex of 3–25 Fab bound to postfusion gB. Initial attempts to visualize the complex were complicated by incomplete Fab occupancy, despite the apparent high affinity of the interaction. Deglycosylation of postfusion gB by treatment with Endoglycosidase H (Endo H) improved 3–25 Fab binding (**S3 Fig**) and resulted in homogeneous particles (**Fig 3A**). Two-dimensional (2D) classification of these particles showed fully saturated postfusion gB + 3–25 Fab complexes that exhibited a stoichiometric ratio of three Fabs to a single postfusion gB trimer (**Fig 3B**). These 2D class averages showed that 3–25 Fab binds to the N-terminal of postfusion gB, supporting the results of our epitope-mapping (**Fig 2**). The 2D class averages also displayed a wide range of 3–25 binding angles, suggesting that this domain is flexible and potentially unstructured (**S1 Movie**). These observations agree with previous structural characterizations of postfusion gB, in which crystallization was only possible after removing the flexible N-terminus through trypsin digestion or by truncating the N-terminal 77 amino acids [17]. This conformational flexibility prevented us from generating a single, definitive 3D reconstruction of the gB + 3–25 Fab complex, but allowed us to calculate several 3D volumes that contain density corresponding to the 3–25 Fab at a variety of different positions relative to the postfusion gB density (**Fig 3C**).

To obtain high-resolution information, we pursued crystallographic studies of 3–25 Fab in complex with a 3–25 epitope peptide named gB-p17 (65-HRANETIYNTTLKYG-79). A crystal

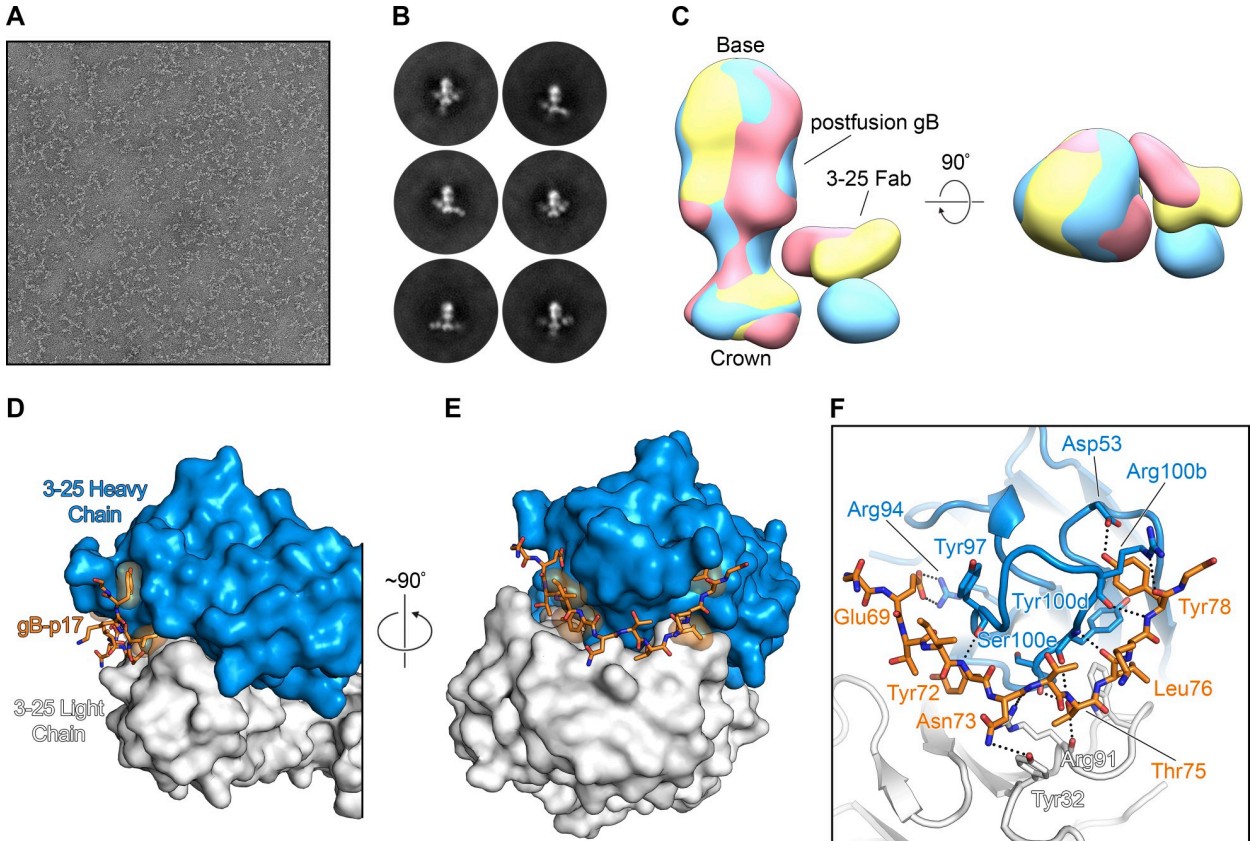

**Fig 3. Structural basis for recognition of postfusion gB by 3–25.** (**A-C**) EM analysis of the postfusion gB + 3–25 Fab complex. (**A**) A representative negative stain micrograph of the postfusion gB + 3–25 Fab complex, collected at a magnification of 92,000×. (**B**) Representative 2D class averages of the postfusion gB + 3–25 Fab complex. (**C**) Three 3D reconstructions of postfusion gB bound by 3–25 Fab are aligned to show the flexibility of the 3–25 epitope. The 3D reconstructions are aligned to one another based on the position of the postfusion gB density and are colored yellow, blue, and red. The crown end of postfusion gB containing 3–25 Fab binding sites and the base end of postfusion gB containing fusion loops are indicated. (**D-F**) The crystal structure of 3–25 Fab bound to gB-p17 peptide. (**D-E**) The 3–25 Fab is shown as a molecular surface, with the heavy chain in blue and the light chain in white. The gB-p17 peptide is shown as orange sticks, with residues that form hydrophobic contacts with the 3–25 Fab shown as transparent orange surfaces. (**F**) The 3–25 Fab is shown as a cartoon with residues that contact the gB-p17 peptide shown as sticks. Hydrogen bonds and salt bridges are shown as black dotted lines.

in space group *C*2 with a single complex per asymmetric unit diffracted X-rays to a resolution of 1.8 Å. After molecular replacement and manual building, the structure was refined to an $R_{work}/R_{free}$ of 16.0%/18.6% (**S3 Table**). This high-resolution structure revealed an extensive hydrogen-bonding network beginning at gB Glu69 and continuing to gB Tyr78, which engage complementarity determining regions (CDRs) of both the 3–25 heavy chain and light chain (**Fig 3D–3F**). Three amino acids (His65, Arg66, Ala67) at the N-terminus of gB-p17 were not resolved in our crystal structure, presumably because these residues were flexible, having not been productively engaged by 3–25 Fab. These three residues, which lie outside of the 3–25 core epitope, are the amino acids that vary among the HCMV strains used in neutralization assays (**S2 Table**). However, all HCMV strains were efficiently neutralized by 3–25 (**S1 Fig**, **S1 Table**), suggesting these residues are not critical for virus neutralization by 3–25. Asn73, located near the center of gB-p17, forms a hydrogen bond with Tyr32 from 3–25 light chain CDR1 (**Fig 3F**). Asn73 of gB is also an *N*-linked glycosylation site based on sequence prediction (**S2 Table**). Unlike the gB-p17 peptide, presence of a branched glycan at Asn73 of gB protein may cause steric hindrance for 3–25 binding, which partially explains glycosylation

sensitive binding of 3–25 to postfusion gB (**S3 Fig**). Intriguingly, the N73A substitution in our epitope-mapping experiments resulted in a substantial increase in relative binding potency (**Fig 2D**), although it should be noted that these experiments were also carried out using peptides that lacked glycosylation. Other detailed interactions between 3–25 and gB-p17 include: Glu69 of gB-p17 forms a salt bridge with Arg94 of 3–25 heavy chain CDR3, anchoring the N-terminus of the peptide; Thr75 of gB-p17 forms hydrogen bonds with Arg91 of 3–25 light chain CDR3, and Tyr100d and Ser100e of 3–25 heavy chain CDR3; Tyr78 of gB-p17 is tucked into a pocket formed by Arg100b of 3–25 heavy chain CDR3 and forms hydrogen bonds with the backbone of Arg100b and the side chain of Asp53, which is from 3–25 heavy chain CDR2 (**Fig 3F**). These detailed interactions between 3–25 and its epitope revealed by the crystal structure are consistent with our epitope mapping results and also help to explain broad HCMV neutralizing ability of 3–25.

## Inhibition of HCMV entry at a post-attachment step by 3–25

Cellular attachment is the first step of viral infection. Inhibition of virus attachment by high-affinity antibodies binding to key viral proteins is a major mechanism of antibody-mediated virus neutralization. The gB protein showed a nanomolar binding avidity ($1.74 \pm 0.06$ nM) to captured 3–25 IgG as determined by bio-layer interferometry (BLI) assay (**Fig 4A**). The 3–25 Fab showed a high binding affinity ($48.8 \pm 0.76$ nM) to the captured epitope peptide gB $P_{064-083}$ but poor binding to the captured gB protein (**S4 Fig**). To determine whether 3–25 recognizes native gB, MRC-5 cells infected with AD169rev (MOI = 1.0) were stained with 3–25 or control IgG and analyzed by flow cytometry assay. As shown in **Fig 4B**, 3–25 generated a stronger signal (blue histogram) compared to that of control antibody (red histogram), suggesting binding of 3–25 to gB expressed on infected cells. To determine whether 3–25 inhibits HCMV attachment, AD169rev was pre-incubated with or without antibodies or with heparin before being applied to target cells at low temperature ($4°C$), which allows viral attachment but prevents internalization and membrane fusion. Detection of cell-attached virus by Western blot revealed that the signals of viral pp65 and gH proteins were comparable for cells attached with virus-only control and for cells attached with virus mixed with 10-fold decreasing concentrations of 3–25 or control IgG. This phenomenon was observed in both ARPE-19 cells and MRC-5 cells. In contrast, a dose-dependent reduction of viral proteins pp65 and gH was detected when the virus was pre-incubated with heparin (**Fig 4C**), which is consistent with previous reports that the initiation of HCMV infection requires an initial interaction with cell-surface heparan sulfate [35]. These results demonstrated that 3–25 does not significantly affect HCMV attachment.

Next we assessed the ability of 3–25 to inhibit viral infection at a post-attachment step following the protocol shown in **Fig 4D**. Briefly, AD169rev-GFP was attached to ARPE-19 cells at $4°C$. After culturing the virus-attached cells at $37°C$ for varying lengths of time, antibodies were added to the cells. The antibody-containing medium was replaced by fresh medium 1 h later and infection was examined 48 h later. When the virus-attached cells were cultured at $37°C$ for 0 min, 3–25 efficiently inhibited HCMV infection of ARPE-19 cells with an $IC_{50}$ of 9.65 nM (1.45 μg/mL), while the isotype control antibody did not show significant viral inhibition (**Fig 4E**). To determine the time window for 3–25 to inhibit virus at a post-attachment step, saturated 3–25 antibody (10 μg/mL) was added to virus-attached ARPE-19 cells at different time points (0, 20, 40 and 60 min) after culturing at $37°C$. As shown in **Fig 4F**, the percentage of viral inhibition decreased as the time before addition of 3–25 increased. The percentage of viral inhibition dropped more than 50% when 3–25 antibody was added at 60 min compared to 0 min. Surprisingly, 3–25 at the same concentrations showed significantly lower virus

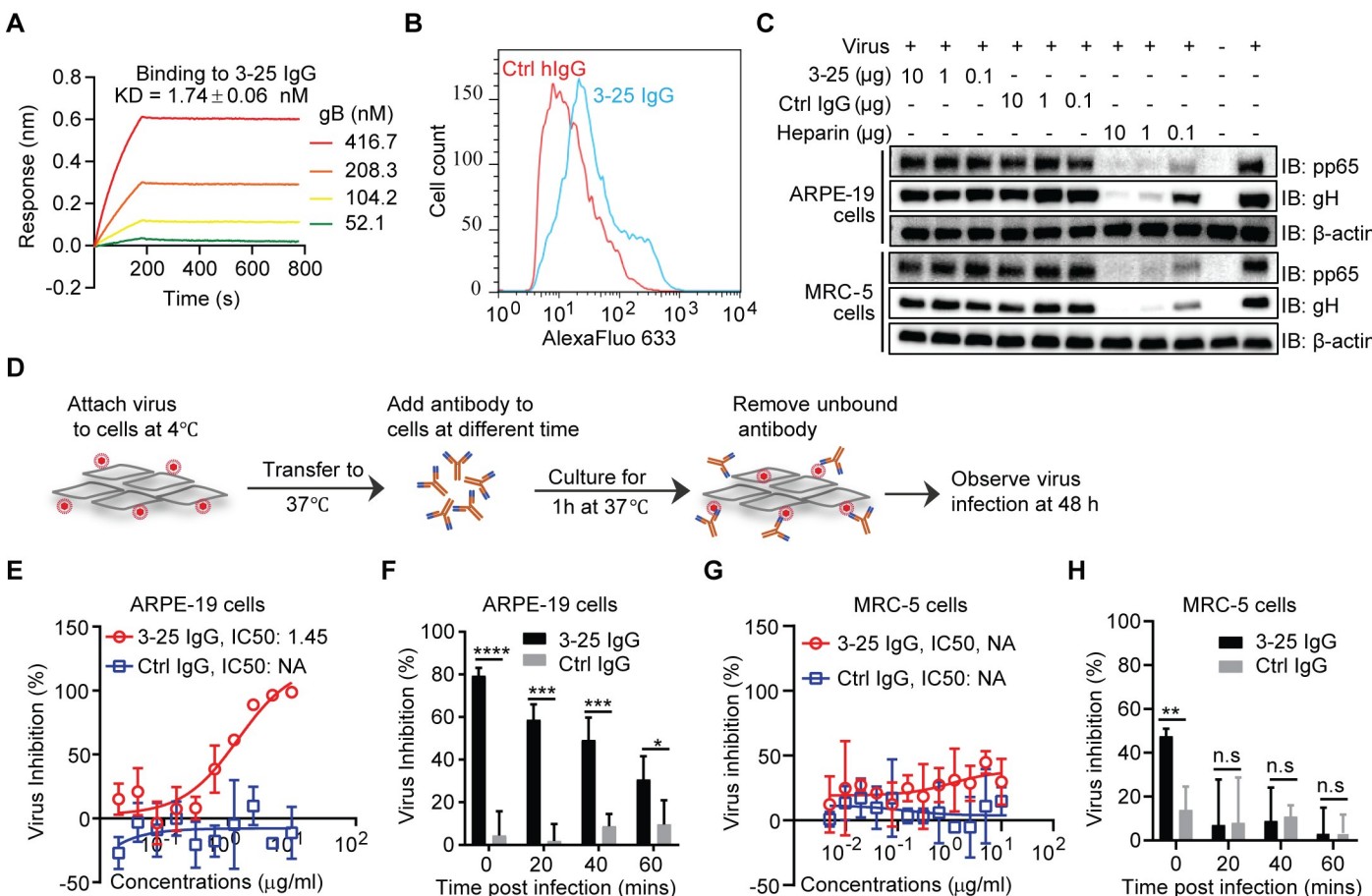

**Fig 4. 3–25 inhibits HCMV infection at a post-attachment step.** (**A**) Binding avidity of recombinant gB to 3–25 captured on Protein A sensors determined by bio-layer interferometry (BLI) assay. (**B**) Detection of gB on surface of AD169rev-infected MRC-5 cells by FACS. (**C**) Virus attachment inhibition. AD169rev was pre-incubated with 3–25, control IgG, or heparin in 200 μL medium and then attached to pre-cooled MRC-5 cells or ARPE-19 cells at 4°C for 1 h. After removing unbound virus, the cell-attached virus was detected by anti-gH and anti-pp65 antibodies by Western blot. β-actin served as a loading control. (**D**) A diagram of the HCMV post-attachment assay. (**E**) Titration of antibodies inhibiting HCMV infection at a post-attachment step in ARPE-19 cells as in (D) when antibodies were added at 0 min. (**F**) Inhibition of cell-attached virus when 10 μg/mL of antibodies were added at different times (0, 20, 40, and 60 min). (**G**) Titration of antibodies for inhibiting HCMV infection at a post-attachment step when antibodies were added at 0 min and (**H**) the inhibition of cell-attached virus when antibodies were added at different times (0, 20, 40, and 60 min) at a post-attachment step in MRC-5 cells.

inhibition in MRC-5 cells than in ARPE-19 cells at a post-attachment step (**Fig 4G**). Significant virus inhibition by 3–25 in MRC-5 cells was detected only when the antibody was added at 0 min (**Fig 4H**). These results demonstrated that 3–25 is capable of preventing HCMV infection at the post-attachment step. The discrepancy of virus inhibition by 3–25 in ARPE-19 cells and MRC-5 cells at the post-attachment step may be related to different HCMV entry routes in these two cell types [36].

A crucial step of enveloped virus infection is membrane fusion with target cells, which allows release of the viral genome containing capsid into the cytosol. Shortly after HCMV entry, the major tegument protein pp65, which is tightly enclosed by the viral envelope, is released from intact virions and relocates to the nucleus independent of the viral capsid [37]. Thus, pp65 nuclear translocation can serve as a marker for successful membrane fusion during HCMV infection. MRC-5 cells grown on chamber slides were incubated with AD169rev (MOI = 10) at 4°C for 1 h. After removing unbound virus, antibodies were added to the cells and cultures were incubated for 5 mins or 3 h at 37°C. Cells were then stained for pp65

(green), early endosome marker EEA1 (red), and nucleus (To-pro-3, blue). A non-neutralizing gB-binding antibody, named 1–155, was included as control [23]. As shown in **Fig 5**, pp65-specific signals were detected in HCMV-infected cells but not in mock-infected cells. The signals for pp65 were very weak and were detected as tiny puncta around the edge of HCMV-infected cells treated with 3–25, 1–155 or medium only at 5 min, which is consistent with intact HCMV virions associated with the cells. Strong signals of pp65 were detected exclusively in the nucleus of 1–155 and medium-only treated cells at 3 h, suggesting successful viral membrane fusion and nuclear translocation of pp65 in these cells. In contrast, the pp65 signals in 3–25 treated cells were mostly detected as bright puncta at perinuclear regions at 3 h, suggesting abortive virus infection with trapped virions that failed to accomplish membrane fusion. Similarly, viral membrane fusion and pp65 nuclear translocation were efficiently inhibited in ARPE-19 cells by 3–25 antibody treatment (**S5 Fig**). These results demonstrate that 3–25 inhibits HCMV infection at a post-attachment step by interfering with viral membrane fusion with target cells.

## Inhibition of post-infection viral spreading by 3–25

Glycoprotein B is strictly required for HCMV entry and cell-to-cell spread as demonstrated by a gB-null virus [18]. Here we determined whether 3–25 inhibits post-infection HCMV shedding and spreading. Confluent ARPE-19 cells were infected with AD169rev-GFP at a very low MOI for three days. Then, the virus-containing medium was replaced with fresh medium with or without antibodies. At 12 days post infection, HCMV infection was detected through GFP expression. As shown in **Fig 6A**, bright GFP-positive plaques were detected in virus-infected cells but not mock-infected cells. Fewer and smaller GFP-positive viral plaques were detected in 3-25-treated cells than in control antibody treated cells (**Fig 6A and 6B**). The sizes of viral plaques, as indicated by GFP signal, were consistent with those observed under bright-field microscopy (**Fig 6B**). Quantitation of GFP-positive viral plaques revealed that the number of viral plaques in 3-25-treated cells was about 65% of that in control IgG-treated cells and virus-only cells (**Fig 6C**). The average size of GFP-positive plaques in 3-25-treated cells was about 43% of that in control IgG and virus-only cells (**Fig 6D**). These results demonstrated that 3–25 restricted HCMV infection even when added at three days post infection. The reduced number and smaller sizes of GFP-positive viral plaques by 3–25 treatment suggest that 3–25 inhibits infectious virus release and cell-to-cell spread post HCMV infection.

## Bivalent-binding-dependent neutralization activity of 3–25

Our results demonstrated that a single postfusion gB trimer was occupied by three 3–25 Fabs (**Fig 3**). This prompted us to investigate whether bivalency of 3–25 plays a role in HCMV neutralization. We did a side-by-side characterization of binding and neutralizing ability of the monovalent 3–25 Fab, bivalent 3–25 (Fab)$_2$, and bivalent 3–25 IgG. As shown in **Fig 7A**, 3–25 Fab was generated through digestion of 3–25 IgG with papain, which cleaves the antibody at the upper hinge region [38]. The 3–25 (Fab)$_2$ was generated by digestion of 3–25 IgG with IdeS enzyme, which cleaves the antibody at the lower hinge region [39]. The sizes and purity of 3–25 Fab, (Fab)$_2$, and IgG were confirmed by SDS-PAGE and Coomassie blue staining under non-reducing and reducing conditions (**Fig 7B**). As determined by ELISA assay, the EC$_{50}$ values for binding to recombinant gB were 0.33 nM for 3–25 IgG, 0.13 nM for (Fab)$_2$, and 0.29 nM for Fab; the EC$_{50}$ values for binding to whole virion were 18.9 nM for 3–25 IgG, 19.5 nM for (Fab)$_2$, and 66.9 nM for Fab (**Fig 7C and 7D**). These results demonstrate that 3–25 (Fab)$_2$ and Fab retained high binding to recombinant gB and whole virions which was comparable to that of 3–25 IgG. Surprisingly, the *in vitro* neutralization assay demonstrated

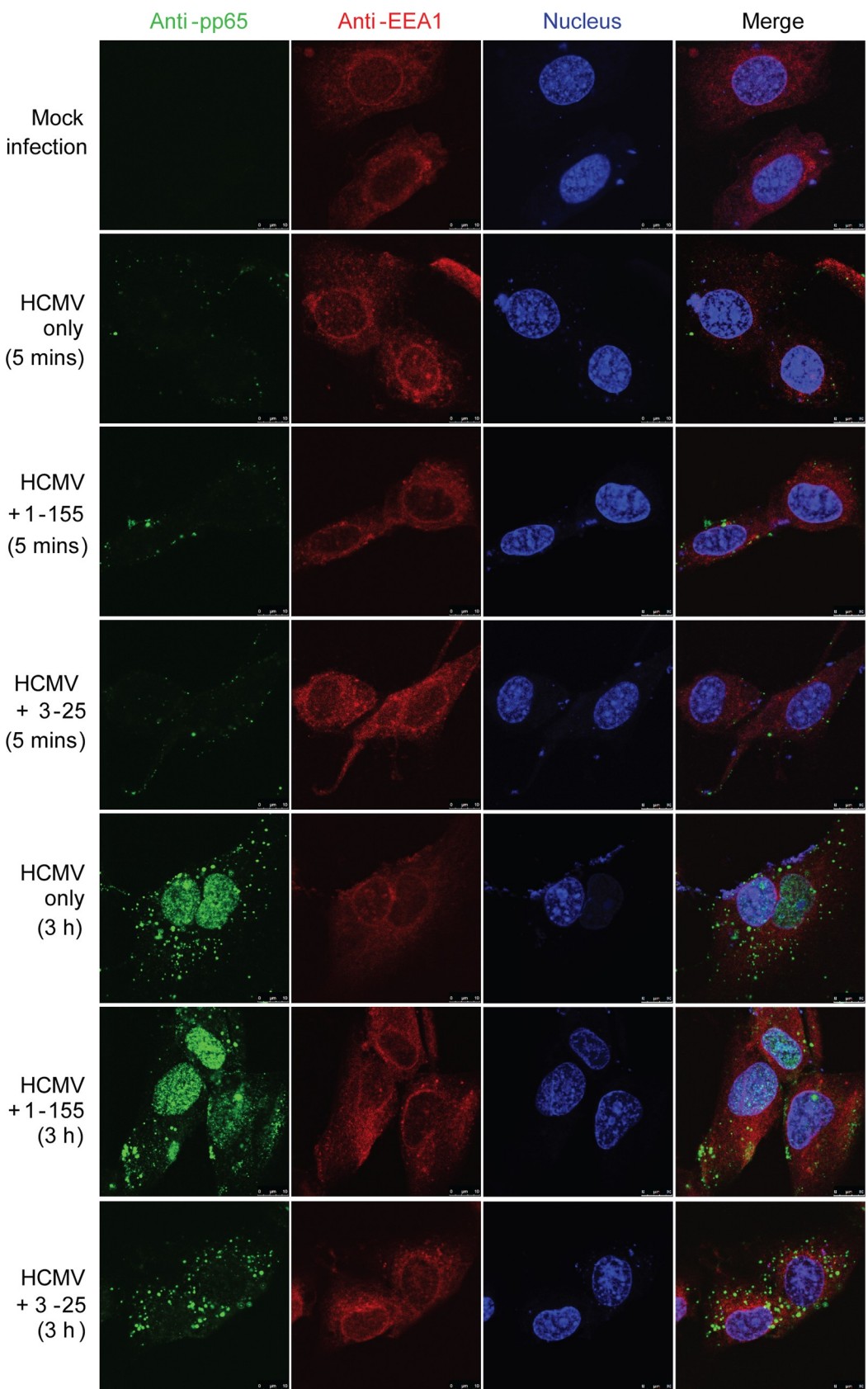

**Fig 5. 3–25 arrests internalized HCMV virions around the perinuclear region in MRC-5 cells.** MRC-5 cells grown in chamber slides were attached with AD169rev at a MOI = 10. After removing unbound virus, 10 µg/mL of 3–25, 1–155 or control IgG was added to the cells and then cultured at 37°C for 5 min or 3 h. The cells were fixed, permeabilized, blocked, and double stained with mouse anti-pp65 and rabbit anti-EEA1 antibodies, and corresponding fluorescently labelled secondary antibodies. Nuclei were stained with To-pro-3 (blue). Bar = 10 µm.

that 3–25 Fab lost activity in both ARPE-19 cells and MRC-5 cells, showing only about 20% virus inhibition at the highest concentration tested (50 µg/mL). In contrast, virus neutralization by 3–25 (Fab)$_2$ was comparable to that of 3–25 IgG (**Fig 7E and 7F**). Together, these results suggested that bivalent binding of 3–25 is required for virus neutralization but not for virus binding. We also characterized the binding and neutralizing ability of another antibody named LJP538, which targets antigenic domain 4 (AD-4) of gB [25]. The EC$_{50}$ values of recombinant gB binding were 0.125 nM for LJP538 IgG, 0.135 nM for (Fab)$_2$, and 0.219 nM for Fab (**Fig 8A**). The IC$_{50}$ for virus neutralization in ARPE-19 cells were 1.71 nM for LJP538 IgG, 0.55 nM for (Fab)$_2$, and 11.7 nM for Fab (**Fig 8B**). Despite recognizing different epitopes, the 3–25 and LJP538 antibodies showed comparable binding and virus neutralizing efficacy. The LJP538 Fab showed robust, albeit reduced, HCMV neutralizing activity compared to LJP538 IgG, indicating that bivalent binding is not absolutely required for HCMV neutralization by LJP538. These data suggested that bivalent-binding-dependent viral neutralization by 3–25 may be attributed to the epitope it recognizes.

## Discussion

The gB/MF59 vaccine demonstrated a moderate efficacy (~50%) in preventing primary HCMV infection in adolescent girls [40] and adult women [41], and reduced viremia in solid organ transplant recipients [42]. However, it seems that gB/MF59 vaccination mainly induced production of nonneutralizing antibodies rather than neutralizing antibodies. Two independent groups reported that the gB/MF59 induced antibodies exhibited very limited *in vitro* virus neutralization activity [43] and minimal inhibiting effect on replication of cell-associated HCMV in a viral spread assay [44]. Consistently, the gB/MF59 induced antibodies showed

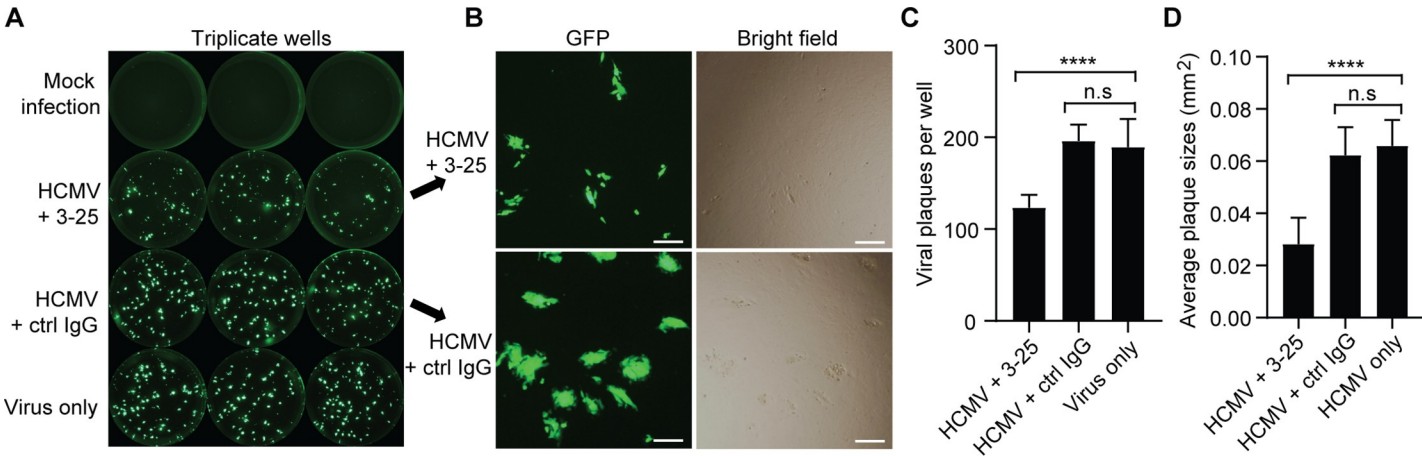

**Fig 6. 3–25 inhibits post-infection HCMV spreading in ARPE-19 cells.** ARPE-19 cells grown in 96-well plates were infected with AD169rev-GFP. At three days post infection, fresh medium containing 10 µg/mL of 3–25 or control IgG or medium only was used to replace the culture medium of infected ARPE-19 cells. At 12 days post viral infection, whole-well GFP images were captured using C.T.L. Immunospot analyzer. (**A**) Representative whole-well images for GFP expression of mock infection, virus control, and infected cells cultured in presence of 3–25 or control IgG. (**B**) Representative pictures showing the typical viral plaques in 3–25 or control IgG treated cells at 12-days post infection. Bar = 50 µm. (**C**) Quantitation of the number of GFP$^+$ viral plaques per well as shown in (A). (**D**) Average sizes of the GFP$^+$ viral plaques per well as shown in (A).

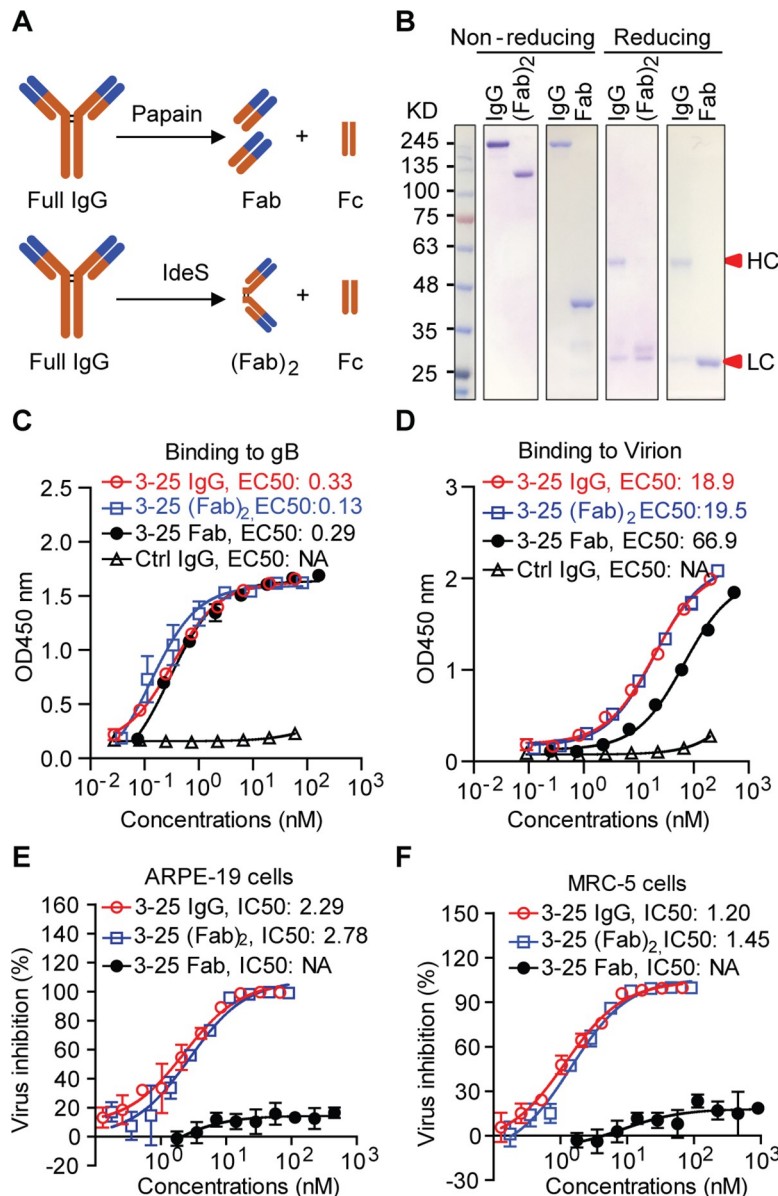

**Fig 7. Bivalent binding is required for virus neutralization by 3–25.** (A) A diagram showing generation of Fab and (Fab)$_2$ from IgG using papain and IdeS enzyme. (B) SDS-PAGE and Coomassie blue staining of purified 3–25 IgG, Fab and (Fab)$_2$, under non-reducing and reducing conditions. (C-D) Comparison of the binding of 3–25 IgG, Fab and (Fab)$_2$ to (C) recombinant gB and (D) whole virion by ELISA assay. The binding EC50 was calculated by non-linear fit of OD450 nm readings vs. concentrations (nM). (E-F) Comparison of the neutralization ability of 3–25 IgG, Fab and (Fab)$_2$ against AD169rev-GFP infection in (E) ARPE-19 cells and (F) MRC-5 cells. The IC$_{50}$ was calculated by non-linear fit of virus inhibition % vs. antibody concentration (nM).

minimal responses against gB structural motifs targeted by neutralizing antibodies including AD-1, AD-2 and structural domain I [43, 44], which is distinct from that of gB-specific antibodies induced by natural infection. Thus, enhancing of induction of neutralizing antibodies is one way to improve vaccine efficacy. Knowledge of the structure of important neutralizing antibodies in complex with gB provides insight into neutralization mechanisms and facilitates development of structure-based design of HCMV vaccines. Structural basis for gB recognition

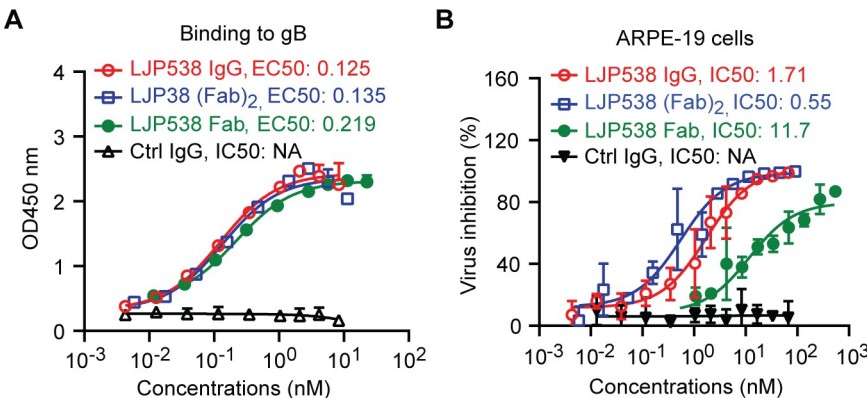

**Fig 8. Bivalent binding is not required for HCMV neutralization by LJP538 antibody. (A)** The binding of LJP538 IgG, Fab and (Fab)$_2$ to recombinant gB determined by ELISA assay. (**B**) The neutralization ability of 3–25 IgG, Fab and (Fab)$_2$ against AD169rev-GFP infection in ARPE-19 cells. The binding EC$_{50}$ or IC$_{50}$ was calculated by non-linear fit of OD450nm readings or virus inhibition % vs. antibody concentration (nM) using GraphPad Prism 5 software.

and virus neutralization by an AD-4 specific neutralizing antibody (named SM5-1) [45] and an AD-5 specific neutralizing antibody (named 1G2) [46] have been implicated by co-crystal structures of these Fabs in complex with either gB domain II or postfusion gB. In this study, we performed a comprehensive characterization of a potent and broadly neutralizing human antibody named 3–25, which targets AD-2 site I of gB. We report a 1.8 Å crystal structure of 3–25 Fab in complex with its epitope peptide and negative-stain EM 3D reconstructions of 3–25 Fab bound to postfusion gB. Our studies revealed the molecular determinants of 3–25 binding to gB at atomic resolution and showed that 3–25 Fab fully occupied trimeric postfusion gB at the N-terminus with flexible binding angles.

Our initial attempts to visualize the postfusion gB trimer in complex with 3–25 Fab revealed dramatic conformational flexibility at the N-terminus of gB and the ability of 3–25 Fab to adopt a wide variety of binding angles. Previous structural characterizations of the postfusion gB trimer by X-ray crystallography [17, 46] were forced to exclude this domain, which supports the hypothesis that AD-2 is both flexible and unstructured in the postfusion conformation. Crystallographic analysis of 3–25 Fab in complex with its epitope peptide elucidated the molecular determinants of binding at atomic resolution, revealing an extensive hydrogen bonding network between the highly conserved residues 69–78 of gB AD-2 and the CDRs of 3–25. This structure confirms our epitope-mapping results and it provides an explanation as to why 3–25 is capable of potently neutralizing such a wide variety of HCMV strains. Interestingly, 3–25 Fab showed glycosylation-sensitive binding to postfusion gB. Glycosylation at the four predicted N-linked glycosylation sites (Asn37, Asn68, Asn73 and Asn85) around or within the 3–25 epitope may reduce antibody binding to this epitope. This reduced binding, could, in turn, increase virus resistance to neutralizing antibodies like 3–25. We and others have shown that only a small group of HCMV-infected individuals develop AD-2-specific antibody responses [47]. The gB/MF59 vaccination produced poor antibody responses against the AD-2 region [43, 44]. Extensive glycosylation at the N-terminus of gB [17] may cause diminished immunogenicity of this highly conserved epitope. AD-2 site I was identified as a linear epitope. However, the peptide-based vaccine cannot elicit potent neutralizing antibodies such as those isolated from HCMV infected individuals. Preclinical evaluation showed that the AD-2 peptide-conjugate vaccines induced only weakly neutralizing serological responses despite robust antibody binding titers [48]. Thus, the structure of the AD-2 region in gB/MF59 may be different from that of native gB, which cannot be simply recreated by linear peptides. A

structure-based optimization of gB antigen may be needed to efficiently induce production of AD-2 targeted potently neutralizing antibodies.

We previously reported that complement could enhance *in vitro* neutralizing potency of some non-neutralizing gB binders and immune sera induced by gB/MF59 vaccination [49]. Here our AD-2 site I specific antibody 3–25 demonstrated potent and broad HCMV neutralization ability without complement in two cell lines. The neutralizing activity of other reported AD-2 site I specific antibodies, including TRL345 and ITC88, also did not rely on complement [50, 51]. Thus, the virus neutralizing ability of AD-2 site I specific antibodies does not dependent on complement. Consistent with a previous report that gB is not required for HCMV attachment [18], 3–25 antibody did not prevent HCMV attachment although it had a high binding avidity to recombinant gB and recognized gB expressed on infected cells. Rather, 3–25 efficiently inhibited HCMV infection at a post-attachment step and showed time-dependent viral inhibition when added at different time points after infection. HCMV major tegument protein pp65 is translocated to the nucleus independent of the viral capsid shortly after HCMV entry [37]. As pp65 is tightly enclosed by the viral envelope, nuclear translocation of pp65 after HCMV exposure can serve as a marker for successful viral infection and membrane fusion with target cells. Treatment of virus-attached cells with 3–25 efficiently prevented HCMV viral membrane fusion and nuclear translocation of pp65. In addition, 3–25 antibody significantly reduced progeny virus release and cell-to-cell spreading in ARPE-19 cells when added at three days post infection. Inhibition of viral membrane fusion and post-infection viral spreading by AD-2 site I specific antibodies is also supported by previous studies [50, 51]. Thus, inhibition of HCMV infection at a post-attachment step through interfering with viral membrane fusion may be a common mechanism for AD-2 site I specific neutralizing antibodies. Although several AD-2 site I specific neutralizing antibodies have been reported, there was no mention of virus escape mutants. Future studies for generation of viral mutants at this highly conserved epitope are needed to assess the role of AD-2 in HCMV gB-mediated viral membrane fusion.

Bivalent-binding-dependent or -enhanced virus neutralization by antibodies has been reported previously in other enveloped viruses. A dengue virus-1 (DENV-1) DIII-specific therapeutic antibody E106 neutralizes DENV-1 infection through bivalent engagement of adjacent DIII subunits on a single virion [52]. Bivalent binding also contributes to HIV-1 neutralization by monoclonal antibodies 2F5, b12, and 4E10 [53, 54]. We showed that 3–25 depends on bivalent binding for HCMV neutralization but not for virion binding. In contrast, gB AD-4 specific antibody LJP538 [25] did not require bivalent binding for HCMV neutralization. The Fc region did not contribute significantly to viral neutralization because the bivalent 3–25 (Fab)$_2$ showed neutralization potency comparable to that of 3–25 IgG. An early study reported that the bivalent format scFv of the gB AD-2 specific antibody ITC-88, but not its monovalent scFv, neutralized HCMV infection [55]. It is possible that bivalency is a common requirement for HCMV neutralization by gB AD-2 specific antibodies. Bivalency of 3–25 may contribute to HCMV neutralization through 1) increased binding avidity of 3–25 antibody and 2) cross-linking of adjacent gB subunits. We and others have demonstrated that the AD-2 specific neutralizing antibodies function before viral membrane fusion with the target cell. The prefusion gB structure of herpesvirus (including HCMV) is still not available, which leaves a major gap in our knowledge about the mechanism of herpesviruses membrane fusion. The postfusion gB structure of HCMV resembles the postfusion structure of HSV-1 and EBV homologues [17]. The epitope of 3–25 is located at the N-terminal region of HCMV gB. Due to high flexibility, the N-terminal region is unresolved in all currently available gB structures [17]. The N-terminal region of HSV-1 gB is also targeted by neutralizing antibodies, but the essential function of this region for virus entry is still unknown [56]. Results of a study suggest domain II of HSV-1

gB is involved in interacting with gH/gL and is essential for viral membrane fusion [57]. Two putative hydrophobic fusion loops are located at domain I of HCMV gB [58]. The 3–25 epitope is located at the crown end of postfusion gB, which is distant from domain II that is in the center of gB and the fusion loops containing domain I that is at the opposite base end of gB. Thus, it is unlikely that 3–25 can directly inhibit membrane insertion of fusion loops or gB interaction with gH/gL. We propose that the 3–25 antibody binds the N-terminal region on both prefusion gB and postfusion gB. Bivalent binding of 3–25 IgG cross-links adjacent gB subunits to prevent the concerted gB conformational change that is required for viral membrane fusion. A prefusion HCMV gB structure, currently unavailable, would reveal the mechanism of gB-mediated membrane fusion and provide a basis for forming hypotheses about the precise neutralization mechanism of the gB N-terminal region specific antibodies.

In summary, we have determined the structural basis for recognition of postfusion gB by an AD-2 specific HCMV broadly neutralizing antibody, which functions at a post-attachment step through interfering with viral membrane fusion. Bivalent binding of the antibody is required for HCMV neutralization, but not for virus binding.

## Materials and Methods

### Cells, viruses and reagents

ARPE-19 cells (ATCC, CRL-230) and MRC-5 cells (ATCC, CCL-171) were cultured as described previously [59]. Three genetically modified laboratory-adapted HCMV strains (Towne-ts15-rR, AD169rev and AD169rev-GFP) with repaired pentamer expression and epithelial tropism were used in this study. The AD169rev-GFP strain contains an in-genome GFP expression cassette. HCMV clinical isolates used for neutralization assays [32] and the gH antibody 223.4 [60] were described previously. Pp65 (Cat#: Ab6503) and EEA1 (Cat#: ab2900) specific antibodies were from Abcam. β-actin antibody (Cat#: A5316) was from Sigma. His-tag antibody was from R&D (Cat#: MAB050). Heparin (Cat#: BP252410) was from Fisher Scientific. The gB peptide library, which contains a total of 169 15-mer peptides (11 amino acids overlapping) covering the extracellular domain of gB (17–703 amino acids of AD169 strain), and biotinylated epitope peptides were purchased from JPT Peptide Technologies GmbH (Berlin, Germany). The numbering of gB sequence in this study is based on numbering used for the AD169 strain.

### Generation of antibodies, Fab and (Fab)$_2$ fragments

Antibodies 3–25 and 1–155 were produced as described previously [23]. The heavy chain and light chain variable region sequences of gB antibody LJP538 [25] were obtained from a published patent (PCT/IB2015/057664) and cloned into human IgG1 backbone for expression. A dengue virus specific human IgG1 antibody [61] was used as isotype control. The 3–25 Fab was generated by digesting 3–25 IgG with papain (Sigma, P4762) as described previously [62]. The 3–25 (Fab)$_2$ was generated by digesting 3–25 IgG with IdeS enzyme (Promega, V7511).

### Generation of postfusion gB

The postfusion gB plasmid encodes residues 32–692 of HCMV gB (strain AD169) with an artificial N-terminal signal sequence and a C-terminal HRV3C protease cleavage site, 8×HisTag, and a TwinStrep tag. The plasmid was transiently transfected into FreeStyle 293-F cells (Gibco) using polyethylenimine for expression. To enhance solubility, we combined published successful engineering strategies of gB expression and introduced the following substitutions: Y155G, I156H, Y157R, Y206H, S238N, W240T, L241T, Y242H and C246S [17, 46]. Cells

transfected with the gB construct were treated with 5 μM kifunensin three hours post-transfection to ensure uniform high-mannose glycosylation. Cell supernatants were harvested six days after transfection. The postfusion gB protein was purified using Strep-Tactin resin (IBA Lifesciences) before being run over a Superose6 10/300 column (GE Healthcare) in 2 mM Tris pH 8.0, 200 mM NaCl, 0.02% $NaN_3$.

### Negative stain EM data collection and processing

The postfusion gB was deglycosylated by digestion with Endo H (10% w/w) for 12 hours at 4˚C and purified using a Superose6 10/300 column. Deglycosylated gB was mixed with a molar excess of 3–25 Fab and run over a Superose6 10/300 column (GE Healthcare) using 2 mM Tris pH 8.0, 200 mM NaCl, 0.02% $NaN_3$. A CF400-Cu grid (Electron Microscopy Sciences) was plasma cleaned for 30 seconds in a Solarus 950 (Gatan) using a 4:1 mixture of $O_2$ to $H_2$. The purified complex was diluted to a concentration of 0.025 mg/mL and mixed with additional 3–25 Fab to fully saturate the gB trimer before being deposited onto the prepared grid. Micrographs were collected using a 200 kV Talos transmission electron microscope (Thermo Fisher) equipped with a Ceta 16M detector (Thermo Fisher) at a magnification of 92,000×, corresponding to a calibrated pixel size of 1.63 Å. Data were collected manually using TIA v4.14 (Thermo Fisher) at a defocus range of 1.8 to 3.1 μm. CTF-estimation, particle picking and preliminary 2D classification were performed using *cis*TEM [63] before particles were exported into cryoSPARC v2.9.0 [64] for *ab initio* model generation and 3D classification.

### Crystallization and structure determination

Purified 3–25 Fab was mixed with a 2.5-fold molar excess of gB-p17 (HRANETIYNTTLKYG) to a final concentration of 12.0 mg/mL. This complex was used for crystallization screens in a sitting drop, vapor-diffusion format. After several days, diffraction-quality crystals appeared in mother liquor composed of 16.75% PEG 400, 13.4% PEG 3350, 0.1 M magnesium chloride, and 0.1 M Tris-HCl pH 8.5. Crystals were cryoprotected by soaking in mother liquor supplemented with 20% glycerol before being plunge frozen into liquid nitrogen. Diffraction data were collected using a Rigaku MicroMax-007HF X-ray generator equipped with a VariMax HighRes detector and were indexed and integrated in iMOSFLM [65] before being merged and scaled to a resolution of 1.80 Å using AIMLESS [66]. A molecular replacement solution was found in PHASER [67] by using a search ensemble generated from PDB IDs: 6BLA and 6DDM. The structure was refined in PHENIX [68] and the gB-p17 peptide was manually built into the resulting electron density map using Coot [69]. All of the crystallographic software programs used in this project were curated by SBGrid [70].

### Virus neutralization assay

Virus neutralization assays with the Towne-ts15-rR, AD169rev, and 12 clinical isolates as shown in Fig 1 were performed in ARPE-19 and MRC-5 cells using an immunostaining method described previously [71]. Neutralization assays with AD169rev-GFP were performed in ARPE-19 cells and MRC-5 cells based on GFP expression of infected cells as described previously [59]. The percentage of virus inhibition by the antibody was calculated. The $IC_{50}$ of antibody was calculated by non-linear fit of virus inhibition % vs. concentration (ng/mL) using GraphPad Prism 5 software. Peptide inhibition assay was performed using AD169rev-GFP. Briefly, 50 μL/well of 10 μg/mL 3–25 antibody (> 50 times of $NT_{50}$) was mixed with 50 μL/well of 2-fold serially diluted 3–25 epitope peptide (64-SHRANETIYNTTLKYGDVVG-83) or non-binding control gB peptide (57-VTSSEAVSHRANETI-71) starting from 25 μg/mL for 30 min at 37˚C, and then mixed with 100 PFU of AD169rev-GFP (50 μL/well) for another

30 min at 37˚C. Cells only, virus only, and 3–25 antibody incubated with virus but no peptide served as controls. The mixtures were added to ~95% confluent ARPE-19 cells or MRC-5 cells grown in 96-well plate. Triplicate wells were determined. Virus infection was quantified 48 h later. Percentage of virus inhibition was calculated. The $IC_{50}$ of peptide to inhibit 3–25 was calculated by non-linear fit of virus inhibition % vs. concentration (ng/mL) using GraphPad Prism 5 software.

### ELISA assay

Individual peptides of gB peptide library (2 μg/mL), recombinant gB (4 μg/mL) or inactivated virion (2 μg/mL) were coated (50 μL/well) in a Costar 96-well high binding plate overnight at 4˚C. Unbound antigens were removed. Following each of the below steps, the plate was washed five times with PBST. The plate was blocked with 200 μL/well of non-fat milk in PBST. For peptide ELISA, 3–25 (2 μg/mL, 100 μL/well) was added to the plates for 90 min at RT and followed by detection with HRP-conjugated goat anti-human IgG (50 μL/well, 1/5000 dilution). For ELISA with gB or virion, 50 μL/well of 2-fold serially diluted 3–25 IgG, Fab or (Fab)$_2$ was added to the plate for 90 minutes at RT and followed by detection with 50 μL/well of 1/5000 diluted HRP conjugated goat anti-human (Fab)$_2$. For the peptide inhibition assay, 50 μL/well of 3–25 antibody at 10 μg/mL was mixed with 50 μL/well of 3-fold serially diluted 3–25 epitope peptide (64-SHRANETIYNTTLKYGDVVG-83) or non-binding control gB peptide (57-VTSSEAVSH-RANETI-71) starting from 9 μg/mL. After incubation at RT for 1 h, the mixtures were added to a plate coated with gB protein or virion for 90 min followed by detection with HRP-conjugated goat anti-human IgG (50 μL/well, 1/5000 dilution). The plate was developed with TMB substrate. Absorbance at 450 nm was recorded on a Molecular Devices Spectra Max M4 machine.

Biotinylated peptides were used in ELISA assay for determination of core epitope and critical amino acid recognized by 3–25. Briefly, 3–25 antibody was coated in the plate (1mg/mL, 100 μL/well) overnight at 4˚C. The plate was blocked and washed to reduce non-specific binding as described above. Serial 3-fold diluted biotinylated peptides were added to 3–25 coated plates and incubated for 90 min. Then, HRP-conjugated streptavidin was added to the plate for 45 minutes. The plate was developed with ADHP (Virolabs, Chantilly, Virginia) for 3–5 min to generate resorufin. The fluorescent signals with excitation at 531 nm and emission at 595 nm were measured (Victor III, Perkin Elmer). The binding $EC_{50}$ of antibody or peptide was determined using GraphPad Prism 5 software.

### Bio-layer interferometry

Protein A biosensors were used to determine the binding affinity between recombinant gB and antibodies 3–25 and 3–155 on an Octet RED 96 system (ForteBio). The biosensors were activated by pre-incubation in kinetics buffer for 20 min. The activated biosensors went through a baseline in kinetics buffer (60 s, 1000 rpm) and a loading step in 10 μg/mL of 3–25 (300 s, 1000 rpm) followed by a second baseline in kinetics buffer (120 s, 1000 rpm), an association step in 2-fold diluted gB protein (10, 5, 2.5, 1.25, 0.625, 0.3125, and 0 μg/mL) in kinetic buffer (180 s, 1000 rpm), and a dissociation step in kinetic buffer (600 s, 1000 rpm). One sensor with no 3–25 loaded was allowed to associate with 10 μg/mL of gB to exclude non-specific binding of antibody on biosensor. One 3–25 loaded sensor associated with kinetic buffer without gB to serve as reference. Data were processed using Octet Data Analysis v10.0 (ForteBio) with a 1:1 binding model.

### HCMV attachment assay

ARPE-19 cells or MRC-5 cells were seeded in 24-well ($1.5 \times 10^5$ cells/well) at 1 day before the experiment. The next day, the cells were pre-cooled at 4˚C for 10 min. $2 \times 10^6$ PFU of

AD169rev was incubated with different amounts of antibody (10 μg, 1 μg and 0.1 μg) in 200 μL medium at 37°C for 30 min. The mixtures were added to the cells before removing old medium from cells, and the plate was incubated at 4°C for 60 min to allow virus attachment. Cells only and cells attached with virus in the absence of antibody were included as controls. The cells were washed 3 times with cold PBS to remove unattached virus and then lysed with 100 μL/well of 2×SDS loading buffer. The samples were boiled at 100°C before SDS-PAGE and Western blot analysis. Viral attachment was detected by pp65-specific mouse antibody (clone 3A12) and gH-specific rabbit antibody (223.4). β-actin was detected by actin-specific mouse antibody and served as a loading control.

## Post-attachment assay

ARPE-19 or MRC-5 cells grown in 96-well plate were pre-cooled at 4°C for 10 min. 50 μL/well of AD169rev-GFP that can generate about 100 GFP positive cells was attached to cells at 4°C for 1 h. The unattached virus was removed by a single wash with cold medium. 50 μL per well of serial 2-fold diluted antibodies (starting from 10 μg/mL) was immediately added to the cells. After culturing for 2 h, medium with antibodies was replaced with fresh medium without antibody. Mock-infected cells and cells infected with virus but not treated with antibodies served as controls. Triplicate wells were determined for each condition. 48 h later, the plate was read using a C.T.L. Immunospot analyzer for quantification of virus infection by GFP expression. The post-attachment kinetic assay was performed and analyzed as above except that 50 μL/well of antibodies at 10 μg/mL were added after culturing AD169rev-GFP attached cells for different time (0 min, 20 min, 40 min and 60 min) at 37°C incubator.

## HCMV spreading assay

ARPE-19 cells grown in 96-well plates were infected with AD169rev-GFP at about 100 PFU/well. At three days post infection, fresh medium containing 10 μg/mL of 3–25 or control IgG or medium only was used to replace the culture medium of infected ARPE-19 cells. At 12 days post viral infection, whole-well GFP images were captured using a C.T.L. Immunospot analyzer as described in the method for virus neutralization assay. The whole-well images for GFP expression were used for quantitation of the number of GFP+ viral plaques per well and the average size of GFP+ viral plaques per well using Image J software. The images showing enlarged viral plaques with GFP expression and bright field were captured using an Olympus fluorescence microscope at 12 days post viral infection.

## Analysis of gB sequences

A total of 317 full-length glycoprotein B protein sequences of HCMV strains were retrieved from the virus pathogen database and analysis resource (ViPR) [34]. The sequences were processed with an in-house script and aligned by multiple sequence alignment (MUSCLE)[72]. 266 sequences with regions around AD-2 motif were analyzed, and frequencies of different sequences between residues 59–88 of gB are summarized in Fig 2E.

## Flow cytometry assay

MRC-5 cells were infected with AD169rev at a MOI = 1.0. The cells were detached using enzyme-free cell disassociation buffer two days later and stained with 10 μg/mL of 3–25 or isotype control antibody diluted in FACS buffer for 1 h on ice. Then, the cells were stained with AlexaFluo633 (AF633) conjugated anti-human IgG (Thermofisher Scientific, A21091) for 45

min on ice. The cells were analyzed using BD FACS Calibur. Mock-infected cells stained with only secondary antibodies were used for gating. Data were processed using FlowJo 4.3.

## Immunofluorescence assay

MRC-5 or ARPE-19 cells grown in chamber slides ($1.5 \times 10^4$ cells/well) were pre-conditioned at 4°C for 10 min. AD169rev was attached to the cells (MOI = 10) by incubation at 4°C for 1 h. Unattached virus was removed from the cells. 200 μL of fresh medium with or without 10 μg/mL of antibody (3–25, 3–155 or isotype control antibody) was added to the cells and cultured at 37°C for 5 min or 4 h. The cells were washed with PBS and fixed with 4% PFA for 20 min at RT. Before each of the following steps, the cells were washed 3 times with PBS. The cells were permeabilized with PBS containing 0.2% NP-40 for 5 min. The cells were incubated with 200 μL per well of PBS with 0.1% BSA, anti-pp65 mouse monoclonal antibody (1/1000 dilution) and anti-EEA1 rabbit polyclonal antibodies (1/500 dilution) at RT for 1 h. Then, the cells were stained with 200 μL per well of PBS with 0.1% BSA, Texas Red-conjugated goat anti-rabbit IgG (1/500 dilution) and AlexaFluor 488 conjugated anti-mouse IgG (Fab)$_2$ (1/2000 dilution) at RT for 45 min. Nuclei were stained with 1/1000 diluted To-Pro3 for 15 min at RT. After 5 times washing with PBS, the chamber slides were sealed under glass cover. Pictures were taken using a Leica confocal microscope.

## Statistical analysis

Statistical significance was determined by unpaired two-tailed Student t-test using Prism 5 software and indicated as follows: n.s., $P > 0.05$; *, $P < 0.05$; **, $P < 0.01$; ***, $P < 0.001$; or ****, $P < 0.0001$. All results are shown as mean values ± standard deviation (SD).

## Supporting information

**S1 Fig.** Twelve clinical HCMV isolates and two laboratory-adapted HCMV strains were used for neutralization assays in (**A**) ARPE-19 cells and (**B**) MRC-5 cells. The IC$_{50}$ was calculated by non-linear fit of the percentage of virus inhibition vs. concentration (ng/mL).
(TIF)

**S2 Fig.** (**A**) Detection of 3–25 epitope-peptide-specific antibody responses in serum samples of 9 HCMV seropositive and 3 HCMV seronegative individuals by ELISA assay. (**B**) 3–25 epitope peptide gB P$_{064-083}$ or a control peptide at different concentrations were pre-incubated with ARPE-19 cells for 1 h before AD169rev-GFP infection. Virus infection as indicated by GFP was quantified using a C.T.L. Immunospot analyzer.
(TIF)

**S3 Fig. 3–25 Fab binding to postfusion gB is influenced by glycosylation.** Sensorgrams showing the binding of 3–25 Fab to glycosylated postfusion gB (left) or Endoglycosidase H (Endo H)-treated postfusion gB (right) are shown. SPR response curves are shown as black lines and the fits used to calculate binding kinetics are shown as red lines.
(TIF)

**S4 Fig.** (**A**) The binding of 3–25 Fab to postfusion gB that was captured on Ni-NTA biosensors and (**B**) the binding of 3–25 Fab to biotinylated epitope peptide gB P$_{064-083}$ that was captured on streptavidin biosensors were determined by bio-layer interferometry (BLI) assay.
(TIF)

**S5 Fig. 3–25 arrests cell-attached virions and inhibits viral membrane fusion in ARPE-19 cells.** ARPE-19 cells grown in chamber slides were attached with AD169rev at a MOI = 10.

After removing unbound virus, 10 μg/mL of 3–25 and 1–155 were added to the cells and then cultured at 37°C for 5 min or 3 h. The cells were fixed, permeabilized, blocked, and double stained with mouse anti-pp65 and rabbit anti-EEA1 antibodies, and corresponding fluorescently labelled secondary antibodies. Nuclei were stained with To-pro-3 (blue). Bar = 10 μm.
(TIF)

**S1 Movie. Flexibility of the 3–25 epitope on postfusion gB.** 2D class averages from negative stain EM were rotated to display postfusion gB in the same orientation and to highlight the dramatic conformational flexibility of the 3–25 epitope.
(MP4)

**S1 Table. IC$_{50}$ of 3–25 and CytoGam against infection of a panel of HCMV isolates in ARPE-19 cells and MRC-5 cells.**
(DOCX)

**S2 Table. The gB sequences of clinical isolates used in neutralization assays.**
(DOCX)

**S3 Table. X-ray crystallographic data collection and refinement statistics.**
(DOCX)

## Acknowledgments

We thank Dr. John Ludes-Meyers for assistance with protein expression and purification, Dr. Art Monzingo for assistance with crystallographic data collection. We thank Dr. Georgina Salazar for editing the manuscript.

## Author Contributions

**Conceptualization:** Xiaohua Ye, Tong-Ming Fu, Zhiqiang An.

**Data curation:** Xiaohua Ye, Hang Su, Daniel Wrapp, Daniel C. Freed, Fengsheng Li, Zihao Yuan, Aimin Tang, Jason S. McLellan, Tong-Ming Fu.

**Formal analysis:** Jason S. McLellan, Tong-Ming Fu.

**Funding acquisition:** Zhiqiang An.

**Investigation:** Xiaohua Ye, Hang Su, Daniel Wrapp, Daniel C. Freed.

**Methodology:** Xiaohua Ye, Daniel Wrapp, Daniel C. Freed, Fengsheng Li, Zihao Yuan, Aimin Tang, Jason S. McLellan.

**Resources:** Leike Li, Zhiqiang Ku, Wei Xiong, Dabbu Jaijyan, Hua Zhu, Dai Wang.

**Software:** Zihao Yuan.

**Supervision:** Ningyan Zhang, Tong-Ming Fu, Zhiqiang An.

**Writing – original draft:** Xiaohua Ye, Daniel Wrapp.

**Writing – review & editing:** Xiaohua Ye, Ningyan Zhang, Tong-Ming Fu, Zhiqiang An.

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
