## [Decision Letter · Decision Letter 0]

22 May 2020

Dear Dr. An,

Thank you very much for submitting your manuscript "Structural basis for recognition of glycoprotein B by a bivalent antibody neutralizing HCMV at a post-attachment step" for consideration at PLOS Pathogens. As with all papers reviewed by the journal, your manuscript was reviewed by members of the editorial board and by several independent reviewers. The reviewers appreciated the attention to an important topic. Based on the reviews, we are likely to accept this manuscript for publication, providing that you modify the manuscript according to the review recommendations.

Sincerely,

Robert F. Kalejta

Associate Editor

PLOS Pathogens

Blossom Damania

Section Editor

PLOS Pathogens

Kasturi Haldar

Editor-in-Chief

PLOS Pathogens

orcid.org/0000-0001-5065-158X

Michael Malim

Editor-in-Chief

PLOS Pathogens

orcid.org/0000-0002-7699-2064

Reviewer Comments (if any, and for reference):

Reviewer's Responses to Questions

**Part I - Summary**

Reviewer #1: This study characterises a monoclonal antibody against gB of HCMV identifying the epitope, demonstrating anti-HCMV activity in vitro argue that it works post binding to neutralise infection and also can limit cell to cell spread in a multi-step growth assay. Finally, they suggest that it works via bi-valent binding although no mechanistic data to support this is divalent binding is required for activity is provided. Overall the experiments are generally well executed and presented fairly although some clarifications are needed in certain areas

Reviewer #2: This paper by Ye et al. presents a functional analysis of 3-25, a human monoclonal antibody recognizing the HCMV fusion protein gB. The authors show that 3-25 has a broad and potent neutralizing activity against 14 HCMV strains. It binds to a specific epitope within AD-2, which is highly conserved among all known HCMV gB proteins. Binding to this epitope was also confirmed in structural studies with 3D reconstruction. By binding to gB, 3-25 blocks the fusion of the viral envelope with cellular membranes and inhibits gB-dependent cell-to-cell spread. Finally, the authors show that bivalent binding of 3-25 to gB is required for neutralization.

Overall, the data are very clear and convincing, and the paper is well written. This study is the most detailed and comprehensive analysis of an HCMV-neutralizing antibody I have seen in a single paper, and this is also the greatest strength of the study. A weakness is that most of the analyses and key findings are not really novel. Co-crystallizations of gB peptides with neutralizing antibody fragments have been done before, and the requirement of bivalent binding for neutralizing antibodies directed against AD-2 has also been shown before. Nevertheless, the beautiful and comprehensive analysis of a potent gB-neutralizing antibody with great potential for clinical application make this study a valuable contribution to the field.

Reviewer #3: In this manuscript, Ye et al reported the functional, biochemical and structural characterization of an antibody, 3-25, isolated from a HCMV seropositive donor. 3-25 potently neutralizes epithelial and fibroblast cells infection by 14 HCMV strains tested and it appears to require the bivalency in the IgG or (Fab)2 format for activity. Functional analysis further demonstrated that 3-25 does not interfere with HCMV attachment and neutralizes HCMV by preventing the fusion process to occur. Epitope mapping showed that 3-25 targets the conserved site I portion of the antigenic domain 2 (AD2) on HCMV gB. Crystal structure of 3-25 Fab in complex with the gB epitope was determined by X-ray Crystallography at 1.8A resolution. Furthermore, negative-stain EM analysis was used to show the location of the 3-25 binding site on the postfusion conformation of the HCMV gB.

Overall, this manuscript is of adequate quality for publication on PLoS Pathogen. My only major concern is that the AD2 epitope on the prefusion conformation of gB is likely to have a different structural arrangement than on the postfusion conformation, thus the structural data presented here represents an incomplete structural basis for 3-25 neutralization. I note that the authors acknowledge this potential drawback in the discussion.”. Given that the functional characterization (6 figures) outweighs structural data (1 figure), the title should be modified to reflect this.

**Part II – Major Issues: Key Experiments Required for Acceptance**

Reviewer #1: I have no major criticisms that absolutely need addressing experimentally

Reviewer #2: 1. The authors claim to have resolved the neutralizing mechanism of 3-25, as stated explicitly in the running title, keywords, abstract, and elsewhere in the manuscript. However, the underlying molecular mechanism does not seem to be fully explained by the data. Of course, every neutralizing anti-gB antibody must bind to gB and inhibit its main function, i.e. fusion. But how does it inhibit fusion? Does it prevent interaction with gH/gL, membrane insertion, or the conformational switch? A previous paper on a gB-neutralizing antibody used structural data to propose that neutralization occurs by blocking the pre- to postfusion transition of gB (Spindler et al., PLoS Pathog. 2014). The authors need to explain how 3-25 prevents fusion or at least present a plausible model.

2. The authors argue that 3-25 prevents envelope fusion with endosomal membranes. As a result, pp65 is trapped in cytoplasmic dots (presumably endosomes) and does not translocate to the nucleus. In Figures 5 and S4, pp65 does not co-localize with EEA1, a marker for early endosomes. However, the EEA1 staining is quite unusual (why are nuclei EEA1 positive?), raising doubts on the specificity of the EEA1 antibody. More convincing data with a better endosome-specific antibody is needed.

3. The authors use AD159rev for most of their experiments. Please explain why this HCMV strain was used, even though it is known to have an unusual behavior: it induces extensive cell-cell fusion in infected epithelial cells (PMID 16051825). The AD169 gB has recently been shown to be hyperfusogenic and mediate rapid entry into MRC5 fibroblasts (PMID 31427511). Could this explain the "surprising" finding described in lines 221-223? Do the authors see the same phenomenon when they use another HCMV strain?

Reviewer #3: (No Response)

**Part III – Minor Issues: Editorial and Data Presentation Modifications**

Reviewer #1: In Figure 6 the authors investigate cell to cell spread and cite the original gB null paper from Teresa Compton who used classical agarose overlays. I think it is useful to clarify what the authors mean. They are using an AD169 repaired virus so the implication is that it is cell associated not cell free. However, does not a reduction in plaque number also suggest cell free virus under the conditions they are using (no overlay). Have they analysed the amount of cell free versus cell associated virus there are in these cultures? Some comment would be appreciated on the limitations of interpretation

My reading of the paper is that binding of the antibody to the virion is monovalent, neutralisation on the other hand is bivalent – Here, although they sell it as one of the major findings of the paper, I don’t see how important this is without exploring the mechanism. Is neutralisation bivalent as a result of an increase in affinity of the binding or because bivalent binding prevents conformational changes in gB necessary for fusion? Could the authors comment?

They also demonstrate that binding of 3-25 is sensitive to glycosylation (156-158), which may impact neutralisation and efficacy of therapies or vaccines based on anti-AD2 antibodies. However, in spite of incomplete occupancy of the glycosylated form of gB by the Fab, 3-25 Ab neutralises infection. Comment?

Reviewer #2: 4. Considering that several gB-neutralizing monoclonal antibodies have been characterized before, it would have been great to see a side-by-side comparison of 3-25 and its competitors in order to understand whether or not 3-25 has superior neutralizing activity in vitro and has more potential for clinical application. In the present study, the authors used only one previously described antibody as a control in one set of experiments. While a comprehensive experimental comparison may be beyond the scope of the present study, the authors could at least discuss how their antibody compares to the others, based on the data available.

5. Line 101. the statement "100–1000 times lower than the IC50 of CytoGam" is not consistent with the data. 10-200 times lower seems more accurate.

6. Line 36. The general statement that HCMV causes asymptomatic infections in otherwise healthy adults is not true. HCMV infections are often mild and oligosymptomatic, sometimes even asymptomatic.

7. Line 233. "with or without antibody for 5 min or 3 h at 37 °C incubator“ – please correct.

Reviewer #3: In Fig 2D, it will help readers focus on the changes introduced by highlighting the Ala mutations in boldface. In addition, mutations (for example, Y78A etc) can be added to the right of the bars in the bar graph so it is easier to see the effects of corresponding mutations.

In Fig 3C one Fab was shown to bind to one postfusion gB trimer but in the main text the stoichiometry was described as 3:1. Clarifications need to be added to the figure legend. Alternatively, all three Fabs can be shown in Fig 3C. In Fig 3C, it will also be helpful to label gB, 3-25 Fab and the position of fused membrane.

In lines 180-181, the authors refer to the interaction between gB Asn73 and Tyr32 of 3-25 CDRL1 as “salt bridge”, this is not correct since salt bridge occurs between oppositely charged amino acids. Hydrogen bonding is more likely (distance and angle between interacting atoms will confirm the interaction type). If hydrogen bonding cannot be concluded based on distance and angle, “polar interaction” can be used here. The interaction between gB Asn73 and 3-25 Tyr32 seems to be in contradiction to the 20-fold higher binding observed when Asn73 is mutated to Ala (Fig 2D)

The authors reported a binding affinity of ~2nM between 3-25 IgG and postfusion gB; it will be helpful to obtain affinity of 3-25 Fab. To reduce the avidity effect, it might be helpful to conjugate gB on biosensors and use 3-25 Fab as analyte. The binding affinity cited in the main text is different than the KD shown on Fig 4A.

PLOS authors have the option to publish the peer review history of their article (what does this mean?). If published, this will include your full peer review and any attached files.

Reviewer #1: No

Reviewer #2: No

Reviewer #3: No
---

## [Editor Report · Decision Letter 1]

22 Jun 2020

Dear Dr. An,

We are pleased to inform you that your manuscript 'Recognition of a highly conserved glycoprotein B epitope by a bivalent antibody neutralizing HCMV at a post-attachment step' has been provisionally accepted for publication in PLOS Pathogens.

Best regards,

Robert F. Kalejta

Associate Editor

PLOS Pathogens

Blossom Damania

Section Editor

PLOS Pathogens

Kasturi Haldar

Editor-in-Chief

PLOS Pathogens

orcid.org/0000-0001-5065-158X

Michael Malim

Editor-in-Chief

PLOS Pathogens

orcid.org/0000-0002-7699-2064
---

## [Editor Report · Acceptance letter]

24 Jul 2020

Dear Dr. An,

We are delighted to inform you that your manuscript, "Recognition of a highly conserved glycoprotein B epitope by a bivalent antibody neutralizing HCMV at a post-attachment step," has been formally accepted for publication in PLOS Pathogens.

Best regards,

Kasturi Haldar

Editor-in-Chief

PLOS Pathogens

orcid.org/0000-0001-5065-158X

Michael Malim

Editor-in-Chief

PLOS Pathogens

orcid.org/0000-0002-7699-2064